# Circulatory Adipokines and Incretins in Adolescent Idiopathic Scoliosis: A Pilot Study

**DOI:** 10.3390/children9111619

**Published:** 2022-10-25

**Authors:** Emilie Normand, Anita Franco, Nathalie Alos, Stefan Parent, Alain Moreau, Valérie Marcil

**Affiliations:** 1Research Center of the CHU Sainte-Justine, Montreal, QC H3T 1C5, Canada; 2Department of Nutrition, Faculty of Medicine, Université de Montréal, Montreal, QC H3T 1J4, Canada; 3Viscogliosi Laboratory in Molecular Genetics and Musculoskeletal Diseases, Research Center of the CHU Sainte-Justine, Montreal, QC H3T 1C5, Canada; 4Endocrine Service, Department of Pediatrics, CHU Sainte-Justine, Montreal, QC H3T 1J4, Canada; 5Department of Surgery, CHU Sainte-Justine, Montreal, QC H3T 1C5, Canada; 6Department of Biochemistry and Molecular Medicine, Faculty of Medicine, Université de Montréal, Montreal, QC H3T 1J4, Canada; 7Department of Stomatology, Faculty of Dentistry, Université de Montréal, Montreal, QC H3A 1J4, Canada

**Keywords:** adolescent idiopathic scoliosis, adipokines, incretins, resistin, leptin, bone mineral density, metabolism

## Abstract

Adolescent idiopathic scoliosis (AIS) is a three-dimensional malformation of the spine of unknown cause that develops between 10 and 18 years old and affects 2–3% of adolescents, mostly girls. It has been reported that girls with AIS have a taller stature, lower body mass index (BMI), and bone mineral density (BMD) than their peers, but the causes remain unexplained. Energy metabolism discrepancies, including alterations in adipokine and incretin circulatory levels, could influence these parameters and contribute to disease pathophysiology. This pilot study aims to compare the anthropometry, BMD, and metabolic profile of 19 AIS girls to 19 age-matched healthy controls. Collected data include participants’ fasting metabolic profile, anthropometry (measurements and DXA scan), nutritional intake, and physical activity level. AIS girls (14.8 ± 1.7 years, Cobb angle 27 ± 10°), compared to controls (14.8 ± 2.1 years), were leaner (BMI-for-age z-score ± SD: −0.59 ± 0.81 vs. 0.09 ± 1.11, *p* = 0.016; fat percentage: 24.4 ± 5.9 vs. 29.2 ± 7.2%, *p* = 0.036), had lower BMD (total body without head z-score ± SD: −0.6 ± 0.83 vs. 0.23 ± 0.98, *p* = 0.038; femoral neck z-score: −0.54 ± 1.20 vs. 0.59 ± 1.59, *p* = 0.043), but their height was similar. AIS girls had higher adiponectin levels [56 (9–287) vs. 32 (7–74) μg/mL, *p* = 0.005] and lower leptin/adiponectin ratio [0.042 (0.005–0.320) vs. 0.258 (0.024–1.053), *p* = 0.005]. AIS participants with a Cobb angle superior to 25° had higher resistin levels compared to controls [98.2 (12.8–287.2) vs. 32.1 (6.6–73.8), *p* = 0.0013]. This pilot study suggests that adipokines are implicated in AIS development and/or progression, but more work is needed to confirm their role in the disease.

## 1. Introduction

Adolescent idiopathic scoliosis (AIS) is defined as a tridimensional spine deformation of unknown cause that develops between 10 and 18 years old. This type of scoliosis appears in approximately 2–3% of adolescents and is more prevalent among girls [1,2,3,4]. The Cobb angle is used to quantify the spine curvature, and scoliosis is classified as mild (10° to 20°), moderate (21° to 40°), severe (41° to 55°), or very severe (56° or more) [5]. Treatments are tailored according to the patient’s condition and depend primarily on the size, pattern, and severity of the curve, and on the potential of progression based on the child’s remaining growth. Patients are most at risk for curve progression during the growth spurt [6]. Those with curvature 10°–25° with a low risk of progression must be closely monitored by their physician but may never require treatment. Bracing is indicated for growing children with a curvature between 25° and 40° [7]. Surgery is considered for severe cases with a Cobb angle greater than 40° [8]. Untreated scoliosis can progress and cause physical deformities, psychological disturbances, back pain, and, in the worst cases, cardiopulmonary limitations [9,10]. Despite its unknown etiology, it is generally accepted that AIS pathogenesis is multifactorial and includes genetic, biomechanical, neuromuscular, and hormonal factors [11].

In recent years, scattered evidence has suggested that one’s metabolism and energy homeostasis is involved in AIS risk, severity, or progression [12,13,14,15,16,17,18]. Although interesting, this hypothesis has yet been poorly explored. It has been pinpointed that girls with AIS have different anthropometric features than their peers, such as taller stature, lower body mass index (BMI), and systemic low bone mineral density (BMD), but the causes of these discrepancies remain unexplained [19,20,21,22]. Energy metabolism, comprising energy expenditure, appetite regulation, and insulin sensitivity, could explain these differences. These phenomena are regulated by the interplay of several metabolic hormones, including adipokines and incretins, that are also known to influence bone homeostasis and growth [23], thus being potential candidates for AIS etiology.

Adipokines are hormones secreted by the adipose tissue and are well known for their role in glucose metabolism, insulin sensitivity, and inflammation [24,25,26,27,28,29,30,31,32,33,34,35,36,37,38,39,40,41,42,43,44]. Growing evidence supports their implication in bone biology and remodeling [45,46,47,48] (Figure 1). Leptin is an inflammatory adipokine with an anorexigenic function, and some studies found lowered leptin levels in AIS and abnormalities in leptin bioavailability [12,14,16,49]. Adiponectin has the opposite effect of leptin as it is anti-inflammatory, promotes insulin sensitivity, and its serum levels are reduced in obesity [50,51]. Moreover, one study found that AIS patients with osteopenia have higher adiponectin levels compared to those with normal BMD and to healthy controls [52]. Other known adipokines have not yet been studied in AIS, such as resistin and visfatin, but they could also be of interest as they have similar metabolic functions as leptin and adiponectin [53,54,55,56]. While the role of adipokines in AIS pathophysiology is still unclear, available data support altered adipokine pathways and functions in AIS patients.

Incretins, peptides secreted by the gastrointestinal tract in response to nutrient intake, are also involved in these processes [57]. Incretins regulate glucose homeostasis through glucose-induced insulin secretion, enhancement of insulin sensitivity, and inhibition of glucagon release [58]. Accumulating evidence further supports their influence on bone metabolism [59,60]. To this day, only ghrelin has been investigated in AIS, and higher levels were found in AIS patients compared to controls [15,18,61]. However, many other incretins interact together to control energy metabolism, such as glucagon-like peptide (GLP)-1, GLP-2, glucose-dependent insulinotropic polypeptide (GIP), and peptide YY (PYY). Dipeptidyl peptidase-4 (DPP-4) is also of interest as it plays a role in incretin inactivation. In parallel, a metabolomic analysis revealed differential lipid metabolites in AIS patients compared to controls, indicating a disruption in glycerophospholipid, glycerolipid and fatty acid metabolism [62].

We propose that metabolic factors, namely adipokines and incretins, are altered in AIS. We performed a pilot study to compare the anthropometry, body composition, BMD, and metabolic profile of AIS girls with age-matched healthy control girls. We also investigated in both groups the associations between anthropometric parameters, body composition, BMD, and metabolic parameters, including adipokines, incretins, and insulin metabolism.

## 2. Materials and Methods

### 2.1. Study Design and Participant Recruitment

This is a cross-sectional pilot study comparing AIS girls and sex- and age-matched controls. Data on body anthropometry and composition, BMD, nutritional intake, physical activity, and biochemical markers were gathered and compared between the two groups. Inclusion criteria for AIS participants were: having a diagnosis of scoliosis (Cobb angle ≥ 10°) confirmed by the treating physician using clinical and standardized X-ray examination, being aged between 10 and 18 years old, being female, and not having received bracing and/or surgical treatment for scoliosis. Inclusion criteria for control participants were being aged between 10 and 18 years old and female. Exclusion criteria for AIS and control participants were: having a history of deformities, neuromuscular disease, metabolic disease, skeletal dysplasia, connective tissue abnormalities, mental retardation, inflammatory bowel disease (Crohn’s disease and ulcerative colitis), celiac disease, irritable bowel syndrome, diarrhea ≥3 times a day occurring ≥3 months in the last year, blood in stool in most stools, recent steroid intake, being pregnant at the time of enrolment, antibiotic or probiotic intake, and/or acute inflammation within the past month. AIS girls (*n* = 21) were recruited at the scoliosis clinic of the Orthopedic Department at the CHU Sainte-Justine (CHUSJ) in Montreal, Canada. Risser stage, age at diagnosis, Cobb angle, localization, and the number of curves were recorded for each participant. If AIS patients had more than one curve, the major curve (the most severe Cobb angle) was used for analysis. Aged-matched healthy girls (*n* = 20) were recruited via advertisements. Participants were enrolled between January 2017 and December 2018 in a single visit at the time of data collection. The Institutional Review Board of CHUSJ approved the study (approval #2017-1145), and investigations were carried out in accordance with the principles of the Declaration of Helsinki. Written informed consent was obtained from study participants and parents.

### 2.2. Assessment of Anthropometric Parameters and Sexual Maturity

During each participant’s visit, weight, height, and waist circumference were measured. Weight was measured with a calibrated digital scale to the nearest 0.1 kg, while the participant was dressed in light clothes and no shoes. Height was recorded standing and sitting against a wall-mounted stadiometer, without shoes, to the nearest 0.1 cm. For AIS participants, adjusted height was calculated by correcting for trunk loss using Bjure’s formula (Log h = 0.011x + 0.177; where h is the loss of the trunk height caused by scoliosis, and x is the Cobb angle of the major curve) [63]. Waist circumference was measured in the horizontal plane of the superior border of the iliac crest with a tape measure to the nearest 0.1 cm. BMI (weight (kg)/height (m)^2^) and waist-to-height ratio (WHtR, waist circumference (cm)/height (cm)) were calculated for each participant. Corrected height was used in the calculations for AIS participants. Corrected BMI-for-age and corrected height-for-age z-scores were determined with the Anthroplus software v.1.0.4 (Geneva: WHO, 2009, v.1.0.4, http://www.who.int/growthref/tools/en/, last accessed on 26 January 2019) based on the World Health Organization growth charts. To determine the pubertal stage, the Tanner scale (breast development and pubic hair growth) was self-reported by all participants with the help of drawings. The number of months since menarche was also recorded.

### 2.3. Body Composition and Bone Mineral Density

Body composition and BMD were assessed with a dual-energy X-ray absorptiometry (DXA) scan using a Lunar Prodigy densitometer (General Electric Healthcare, Madison, WI, USA) and performed by trained technicians. Areal bone mineral density (g/cm^2^) was measured at three sites: the lumbar spine, the femoral neck, and the total body-less-head, according to The International Society for Clinical Densitometry (ISCD) recommendations in pediatrics [64]. Age-adjusted and sex-adjusted BMD z-scores were calculated with a reference population from the United States (NHANES/Lunar combined; v112) and used for analyses. The DXA scan also provided total body fat percentage (%), lean mass (g), and fat mass (g). The fat mass index and lean mass index were calculated by dividing both by the square of the height (corrected height for AIS participants) [65]. Those indexes normalize the data as height is a confounding factor for body composition parameters.

### 2.4. Assessment of Dietary Intakes

Dietary intakes were evaluated using validated interviewer-administered food frequency questionnaires (FFQ) specific to our population [66]. Measuring cups were used during the interview to estimate portion size. Participants were asked to indicate the consumption frequency of each food item in the last month on a daily, weekly, or monthly basis. Computation of nutrient intakes derived from the FFQs was performed using a validated in-house-built nutrient calculation tool [66]. Intake of vitamin and mineral supplements was also assessed and included in the analysis. Average daily intakes of energy, macronutrients, and micronutrients were calculated for each participant.

### 2.5. Assessment of Physical Activity

Physical activity was assessed using the Canadian Health Measures Survey (CHMS)—Cycle 4 questionnaires, namely the Physical and Sedentary Activity Questionnaires for adolescents (12 to 18 years old) and children (<12 years old). These interviewer-administrated questionnaires record how many hours the participant engaged in moderate to intense physical activities at school and at home in the last seven days and in a normal week. Clarifications were asked about the type, frequency, and intensity of exercises. Average moderate to intense physical activity per day was calculated. Questions on the number of hours of screen time during a typical week over the last three months were also asked to determine participants’ screen time (hrs/day), which included time spent watching television, computer, tablet, and/or phone.

### 2.6. Assessment of Estimated Energy Requirement and Energy Balance

Estimated energy requirement (EER) is the average dietary energy intake predicted to maintain energy balance in healthy, normal-weight individuals of a defined age, sex, weight, height, and level of physical activity consistent with good health [67]. Based on the physical activity questionnaire, participants were classified according to their physical activity coefficients and characterized as sedentary, low active, active, and very active [67]. Energy balance was calculated as energy intake divided by the EER in percentage (energy intake/EER × 100).

### 2.7. Biochemical Determinations

To study participants’ metabolic profile, blood samples were drawn in the morning after a 12-h fasting period. Inhibitors [4-(2-aminoethyl) benzenesulfonyl fluoride hydrochloride (AEBSF; Sigma Aldrich, St-Louis, MO, USA), aprotinin (Roche Diagnostics, Indianapolis, USA) and DPP-4 inhibitor (Millipore, St-Charles, MO, USA) were added to EDTA-treated collection tubes before blood was drawn to maintain the integrity of metabolic molecules. Collected blood was kept on ice until centrifugation. Plasma was separated and stored at −80 °C until analysis.

Total cholesterol, HDL-C, and TG were determined enzymatically with a Synchron LX20 (Beckman Coulter, Brea, CA, USA). The Friedewald equation was used to calculate LDL-C. ApoB was measured by nephelometry (Array Protein System; Beckman) and ApoA1 by an automated analyzer (Cobas Integra 400, Roche Diagnostics). Glucose was measured by the glucose oxidase method, and plasma insulin concentrations were determined using the ultrasensitive Access immunoassay system (Beckman Coulter, Brea, CA, USA).

The homeostasis model assessment insulin resistance (HOMA-IR) was calculated using the formula: insulin (mIU/L) × glucose (mmol/L)/22.5. The quantitative insulin sensitivity check index (QUICKI) was calculated as an indicator of insulin sensitivity with the formula: 1/(log(fasting insulin) + log(fasting glucose).

Plasma adipokines (leptin, adiponectin, resistin), incretins (active ghrelin and GIP), and c-peptide were analyzed with the Milliplex custom Human Metabolic Hormone (HMHEMAG-34K) and the Human Adipokine 1 Magnetic Bead Panel (HADK1MAG-61K)-Metabolic Multiplex Assay (Millipore, St-Charles, MO, USA). Eve Technologies Corporation (Calgary, AB, Canada) performed the experiments. ELISA kits were used to measure levels of total (EZGLP1T-36K) and active GLP-1 (EGLP-35K), GLP-2 (E2GLP2-37K), ghrelin total (EZGRT-89K), and PYY (EZHPYYT66K) (Millipore, St-Charles, MO, USA), as well as visfatin (EH482RB, Thermo Fisher Scientific, Carlsbad, CA, USA), uncarboxylated (446707, Biolegend, San Diego, CA, USA), and total osteocalcin (43-OSNHU-E01, Alpco, Salem, NH, USA). DPP-4 activity was measured using an activity assay kit following the manufacturer’s instructions (BML-AK498, Enzo Life Sciences, Farmingdale, NY, USA).

#### Statistical Analyses

The Kolmogorov-Smirnov test was used to evaluate the normality of distribution. Normally distributed data were reported as mean ± SD, and non-normally distributed data were presented as median (range). Paired for age, Student *t*-tests or Wilcoxon matched-pairs signed rank tests were used to compare control and AIS groups. AIS patients were categorized according to the severity of the curve with the median Cobb angle as the cut-off value (Cobb angle <25° or ≥25°). Levels of adipokines and incretins were compared with Student *t*-tests or Mann-Whitney tests. Associations between adipokines, incretins, or insulin metabolism markers (independent variables) and BMI-for-age, fat percentage, or whole-body BMD were assessed using multiple linear regression analysis. Analyses were performed without covariables (crude models) and with covariables, such as age, fat mass, height-for-age, energy intake, calcium intake, vitamin D intake, and physical activity, when appropriate. *p* < 0.05 was considered statistically significant. Statistical analysis was performed using GraphPad Prism software (version 8.0.1) and IBM-SPSS software (version 25).

## 3. Results

We recruited 20 AIS and 22 healthy control participants (Figure 2). Three girls in the healthy control group were excluded from our analysis: two were diagnosed with dyslipidemia after biochemical assessment, and one was diagnosed with AIS following a DXA scan and was further classified in the AIS group. With 21 AIS and 19 control participants, 19 age-matched pairs were included in the paired analyses. A total of 17 control participants underwent the DXA scan, resulting in 17 age-matched pairs for body composition and BMD analysis. The clinical and anthropometric characteristics of AIS and control participants are detailed in Table 1. Mean age and sexual maturity parameters were similar between groups. AIS participants had a mean Cobb angle of 27.1 ± 10.5° (range: 13–46°) that represents a curve of moderate severity and a mean age at diagnosis of 13.0 ± 1.7 years. Comparing anthropometric features revealed that AIS girls were leaner than controls, as reflected by lower BMI, BMI-for-age z-score, and waist-to-height ratio, but they were not taller.

Participants underwent a full-body DXA scan to precisely determine their BMD and body composition. Table 2 shows that, compared to controls, girls with AIS had lower total body less head and lower femoral neck BMD z-score, but no statistically significant difference was observed at the lumbar spine site. Also, AIS girls had lower total body fat percentage, fat mass, and fat mass index than controls. Lean mass and lean mass index were similar between groups.

Dietary intakes and physical activity levels in both groups are shown in Table 3. The mean caloric intake was similar between groups. There was a trend for superior energy balance in the AIS group, theoretically indicating that they consume more calories than they expend. However, girls with AIS spent significantly less time per week doing moderate-to-intense physical activities than controls and had greater daily screen time.

Fasting glucose and lipid profile were also compared between the two groups (Table 4). Fasting glucose, insulin, HOMA-IR, and QUICKI were not different between groups, and neither were TG, total cholesterol, HDL-C, and ApoA1 levels. There was a tendency for higher LDL-C and ApoB in AIS participants, but the differences were not statistically significant. The ApoB/ApoA1 ratio, a cardiovascular risk marker, was also similar between the AIS and control groups.

Next, we assessed fasting levels of adipokines and incretins (Table 5). Mean levels of circulating visfatin were similar between groups. Leptin and adiponectin were found to be higher in AIS participants compared to controls. This translated into a lower leptin/adiponectin ratio in AIS participants. Also, mean levels of resistin were higher in the AIS group but without reaching statistical significance. After stratification of the AIS group according to scoliosis severity, we found higher levels of adiponectin and resistin in AIS participants with a Cobb angle greater than 25° compared to controls, but no difference in leptin and visfatin levels (Figure 3). Mean levels of fasting incretins (GLP-1 total and active, GLP-2, GIP, ghrelin total and active, and PYY) were not different between groups (Table 5). There was also no difference observed for DPP-4 activity, total, and undercarboxylated osteocalcin between AIS and control groups.

Analyzing incretin levels according to Cobb angle severity did not reveal differences between groups (Appendix A). Similarly, no difference was found when anthropometry, body composition, and BMD were sub-analyzed according to the Cobb angle (Appendix A).

The associations between energy metabolism markers and anthropometry, body composition, and BMD are shown in Appendix A. Adjusted models revealed that leptin was positively associated with BMI-for-age and fat percentage in both AIS and control groups. Similar associations for adiponectin, resistin, and visfatin were not found. Leptin was inversely associated with BMD in the control group but not in AIS. Concerning incretins, total and active ghrelin were inversely associated with BMI-for-age in the AIS group. Also, GIP was inversely associated with whole-body BMD only in AIS participants. In the control group only, total OCN and QUICKI were inversely associated, and HOMA-IR was positively associated with BMI-for-age. Total OCN was inversely associated with fat percentage in the control group. Finally, there was no association between insulin metabolism markers and whole-body BMD.

## 4. Discussion

In this pilot study, we found that AIS girls were leaner but not taller, and had lower total body-less-head and femoral neck BMD compared to age- and sex-matched controls. We also found differences in circulatory levels of adipokines between groups, but fasting incretins were similar. There were associations between insulin metabolism markers (HOMA-IR, QUIKI, and total OCN) and anthropometry in controls but not in the AIS group. These results align with our hypothesis that girls with AIS have a disturbed energy metabolism.

Leptin, an inflammatory adipokine with an anorexigenic function, has been studied in AIS, but its role remains controversial. A meta-analysis of six comparative studies concluded that serum leptin levels between AIS patients and controls were not different [68], while another meta-analysis of seven studies found lower levels in AIS participants compared to controls [17]. The small sample sizes, lack of information on participants’ fasting state, and the different study designs and methods of leptin analysis could explain the contradictory results.

Circulating leptin levels are positively associated with fat mass [69]. A meta-analysis of 3747 participants found that the mean BMI of AIS patients was significantly lower than controls [21]. BMI itself does not accurately reflect adiposity and body composition, especially in children and adolescents [70], but the DXA scan is considered the gold standard for measuring body composition. In our study, considering the observed lower BMI, WHtR, fat mass, and fat percentage in AIS participants, the observed (but not statistically significant) lower leptin levels were expected as they are related to participants’ adiposity. Similarly, the lower adiposity of AIS participants could explain the observed higher adiponectin levels. However, this does not appear to be the case in our study, as no association was found between adiponectin, BMI-for-age, and fat percentage. Corroborating our findings, Zhang et al. found higher plasma adiponectin levels in AIS patients than in controls [52]. However, they also found higher adiponectin levels in AIS patients with osteopenia compared to those with normal bone mass, while no such association was found in our pilot study. In parallel, adiponectin levels have been compared between a group of highly active and less active adolescents [71]. It was shown that girls with higher vigorous physical activity levels had greater adiponectin levels, independently of their weight status [71]. In our study, although girls with AIS spent less time performing moderate to intense physical activity, their adiponectin levels were superior to those of controls. This could indicate disturbances in adipokine production in the AIS group.

To our knowledge, this is the first report examining resistin and visfatin levels in AIS patients. Resistin is associated with obesity and cardiometabolic diseases [72], which is why it was surprising to observe higher levels in the AIS patients with more severe curvatures since they were leaner than controls. Higher resistin levels were also observed in a group of children and adolescents with type 1 diabetes compared to healthy controls [73], supporting that resistin is associated with insulin metabolism discrepancies. The mechanisms and receptors by which resistin exerts its actions are still unknown. It has been proposed that G-protein-coupled receptors are involved in lymphocyte chemotaxis induced by resistin [74]. One can speculate that resistin’s actions on energy and bone metabolism could involve the G-protein-coupled receptor pathway. A signaling dysfunction of G inhibitory proteins was reported in AIS patients [75], where a hypo-functionality of inhibitory Gαi subunits and an enhancement of Gs signaling were correlated with the risk of disease progression. This dysfunction could potentially explain why resistin levels were higher in the AIS group and correlated with disease severity. Furthermore, visfatin is highly present in visceral fat [48], and its circulatory levels were elevated in childhood obesity [76]. Here, no difference in visfatin levels was found between groups.

In addition to adipokines, energy homeostasis is maintained by the actions of incretins. In AIS, ghrelin is the only adipokine studied thus far. It was reported that AIS girls have higher levels of fasting total ghrelin compared to controls [15]. Total ghrelin levels were also higher in AIS patients with progressive curves than stable ones [18]. Also, compared to controls, plasma ghrelin levels were higher in AIS patients with osteopenia [61]. While the active acylated form of ghrelin is known for its metabolic actions and appetite regulation [77], only total ghrelin has been assessed in AIS. Here, no difference between groups was found for either acylated or total ghrelin or any other measured incretins, namely GIP, total and active GLP-1, GLP-2, and PYY. Since incretins are secreted post-prandially, it is possible that the participants’ fasting state prevented the detection of differences between groups. Future studies investigating incretin levels in AIS following a standardized meal could shed some light on energy metabolism regulation in the disease.

We found no difference between groups in the levels of fasting glucose and insulin, as well as in indicators of insulin resistance and insulin sensitivity. However, since associations between HOMA-IR, QUIKI, and BMI-for-age were found only in the control group, one could speculate that AIS patients do not respond to insulin in the same manner as controls. Further studies examining post-prandial insulin response in AIS are needed to validate this hypothesis. Moreover, examining participants’ lipid profiles exposed a tendency for higher LDL-C and ApoB in the AIS group, though lipid levels remained within the normal range. Larger studies are needed to confirm if discrepancies in circulating lipids are present in AIS.

Insulin metabolism is known to influence bone remodeling [78]. In osteoblasts, the activation of the insulin receptor stimulates the expression of OCN, which is also implicated in glucose homeostasis [78,79]. Undercarboxylated OCN is released in circulation, where it stimulates pancreatic β-cell proliferation and increases insulin production, insulin sensitivity, and energy expenditure from peripheral organs [80,81,82]. Here, we found no difference between groups in circulating levels of either form of OCN. However, total OCN was inversely associated with BMI-for-age and fat percentage in the control group but not in AIS. A recent meta-analysis reported that, in humans, higher total OCN levels are inversely correlated with lower BMI and fat percentage. Kulis et al. studied OCN levels in premenarcheal and post-menarcheal AIS and control girls. In contrast to our study, they found higher OCN levels in AIS compared to controls for both menstrual statuses [83]. Matusik et al. studied osteocalcin related to AIS severity [13] and found that osteocalcin was inversely associated with the severity of the scoliotic curve. This association was not found in our pilot study.

In the literature, lower BMD of AIS patients compared to controls has been extensively reported, corroborating our findings [20,22,84,85,86]. Adiponectin and ghrelin could impact BMD in AIS through their actions on the nuclear factor kappa-B ligand (RANKL) and osteoprotegerin (OPG) ratio [61]. In control osteoblasts, treatment with ghrelin increased OPG and decreased RANKL expression in a concentration-dependent manner, which is in favor of decreasing bone resorption. In contrast, there was no significant variation in OPG and RANKL expression in AIS osteoblasts. Treatment of osteoblasts with adiponectin had the opposite effect of ghrelin and rather promoted a higher RANKL/OPG ratio, which leads to an increase in bone resorption [46,52]. Higher adiponectin levels in AIS could reflect a higher RANKL/OPG ratio in osteoblasts, which could impact BMD.

Important factors to consider when studying energy metabolism, anthropometry, and BMD are dietary intake and level of physical activity. As previously reviewed, few studies have investigated nutrition in AIS patients [87]. Furthermore, the literature on physical activity is inconsistent, as some studies found that AIS girls were less active than controls [20,88,89], while others did not observe any differences between groups [16,90,91]. In our pilot study, there was no difference in mean caloric intake between groups. However, AIS girls did less moderate-to-intense physical activity than controls, combined with more hours of screen time. It is known that weight-bearing exercises are important for optimal BMD. In our study, participants in the AIS group did fewer weight-bearing exercises than the controls, which may contribute to their lower BMD. Importantly, both groups met their energy need requirements. Their similar caloric intake and lower physical activity level, BMI, and fat percentage of AIS participants support the hypothesis of a higher metabolism in AIS.

Our pilot study has several strengths and limitations. A major strength is that the study protocol was specifically designed to study metabolic parameters, including a homogenous fasting period, the use of inhibitors, and the collection of dietary and physical activity data. Additional strengths are the use of the DXA scan to assess body composition, the paired study design, and the inclusion of only newly diagnosed patients, thus excluding treatment as a confounding factor. Study limitations include the small sample size, which does not allow for extensive multivariate analysis. Given that this was a pilot and exploratory study, *p*-values were not corrected for multiple testing. While the methods we utilized to measure adipokines and incretins have been used in numerous recent studies [92,93,94,95,96,97], mass spectrometry would provide higher specificity [98,99,100]. Other limits were the study restriction of only females with mild to moderate curve severity and the lack of post-prandial assessments. Future and larger longitudinal studies, including female and male participants, as well as mild and severe cases, are needed to study energy metabolism in AIS thoroughly.

## 5. Conclusions

In conclusion, this pilot study suggests a possible role for adipokines in AIS anthropometric and BMD discrepancies, which could impact disease development and/or progression. Further well-designed larger studies are needed to validate our findings. This research is a first step toward a better understanding of energy metabolism in AIS.

## Figures and Tables

**Figure 1 children-09-01619-f001:**
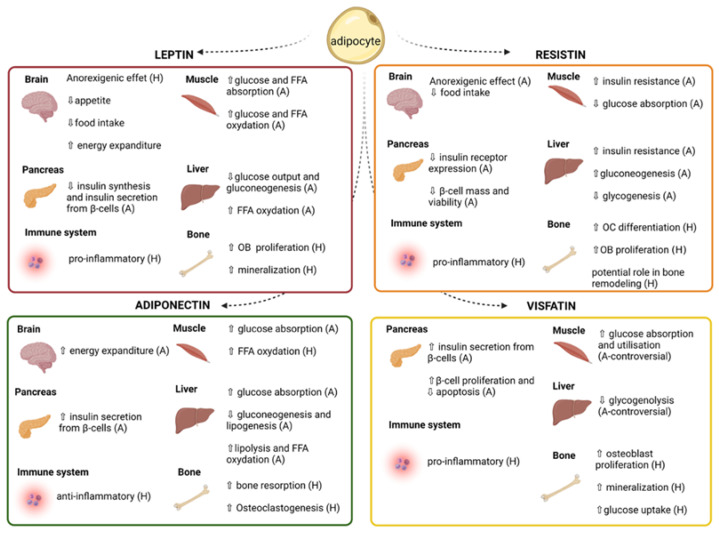
Roles of adipokines on peripheral organs. Leptin, adiponectin, resistin, and visfatin are hormones secreted by the adipose tissue that impact energy metabolism, inflammatory processes, and bone health via their actions on peripheral organs. FFA: Free fatty acid; OB: Osteoblasts; A: Animal studies; H: Human studies. This figure was created using BioRender.com.

**Figure 2 children-09-01619-f002:**
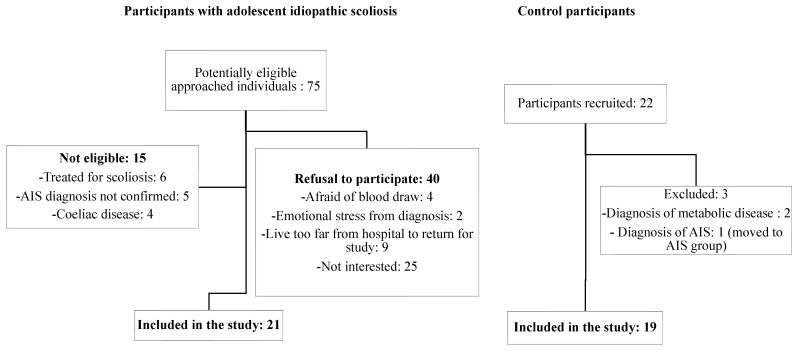
Flow diagram of participant recruitment.

**Figure 3 children-09-01619-f003:**
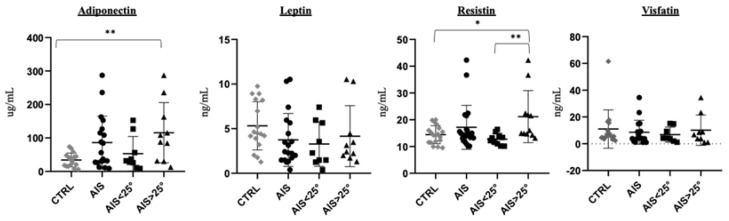
Adipokine levels in control and AIS participants. AIS participants were divided based on curve severity with Cobb angles < 25° or >25° as the cut-off point. Plasma adipokine levels were assessed by multiplex immunoassays in n = 19 CTRL, n = 21 AIS, n = 10 AIS < 25° and n = 11 AIS > 25°. * *p* < 0.05, ** *p* < 0.01 Mann-Whitney test. CTRL: Controls; AIS: Adolescent idiopathic scoliosis.

**Table 1 children-09-01619-t001:** Clinical and anthropometric characteristics of control and AIS participants.

	Controls (n = 19)	AIS (n = 19)	*p* Value ^1^
Family history of AIS (%)	0	63	<0.0001 *
Age (years)	14.8 ± 2.1	14.8 ± 1.7	0.924
Age at diagnosis (years)	-	13.0 ± 1.7	-
Highest Cobb angle (°)	-	27.1 ± 10.5	-
Risser score	-	2.8 ± 1.8	-
Age of menarche (years)	12.0 (9.0–15.0)	12.5 (11.0–14.0)	0.969
Tanner stage (breast)	3.7 ± 1.0	3.4 ± 1.0	0.350
Tanner stage (pubic hair)	3.6 ± 1.1	3.3 ± 0.8	0.310
Height (cm)	160.8 ± 9.9	164.2± 9.3	0.142
Height-for-age (z-score)	0.33 ± 1.01	0.60 ± 1.14	0.419
Weight (kg)	54.2 ± 12.9	50.3 ± 9.0	0.158
BMI (kg/m^2^)	20.6 ± 3.6	18.5 ± 1.9	0.013 *
BMI-for-age (z-score)	0.09 ± 1.11	−0.59 ± 0.81	0.016 *
Waist circumference (cm)	76.0 ± 9.1	71.9 ± 5.6	0.055
Waist-to-height ratio	0.47 ± 0.04	0.44 ± 0.03	0.009 *

BMI: Body mass index; AIS: Adolescent idiopathic scoliosis. Corrected height was used for AIS participants. Data are presented as mean ± SD or median (range). ^1^ Data were compared using a chi-square test, paired *t*-test, or Wilcoxon matched-pairs signed rank test for non-normally distributed variables. * *p* < 0.05.

**Table 2 children-09-01619-t002:** Bone mineral density and body composition of control and AIS participants.

	Controls (n = 17)	AIS (n = 17)	*p* Value ^1^
BMD total body less head (z-score)	0.32 ± 0.88	−0.41 ± 0.80	0.044 *
BMC total body less head (g)	1798.1 ± 414.6	1666.9 ± 293.9	0.286
BMD lumbar spine (z-score)	−0.11 ± 0.87	−0.48 ± 0.93	0.241
BMD femoral neck (z-score)	0.59 ± 1.59	−0.54 ± 1.20	0.043 *
Fat (%)	29.2 ± 7.2	24.4 ± 5.9	0.036 *
Fat mass (g)	15676.1 ± 6106.9	12064.1 ± 3929.8	0.024 *
Fat mass index (g/m^2^)	0.59 ± 0.21	0.44 ± 0.13	0.007 *
Lean mass (g)	36558.5 ± 6466.2	36912.3 ± 5578.1	0.865
Lean mass index (g/m^2^)	1.40 ± 0.18	1.33 ± 0.11	0.273

BMD: Bone mineral density; AIS: Adolescent idiopathic scoliosis. Data are presented as mean ± SD. ^1^ Data were compared using paired *t*-test. * *p* < 0.05.

**Table 3 children-09-01619-t003:** Nutrition and physical activity of control and AIS participants.

	Controls (n = 19)	AIS (n = 19)	*p* Value ^1^
Caloric intake (kcal/day)	2026.6 ± 599.2	2058.1 ± 463.4	0.868
Estimated energy requirement (kcal/day)	2063. 6 ± 341.8	1881.0 ± 221.2	0.066
Energy balance (%)	99.6 ± 28.9	109.1 ± 19.3	0.250
Energy balance (kcal/day)	−36.9 ± 640.6	193.5 ± 360.1	0.178
Moderate-to-intense physical activity (hrs/week)	5.3 (0.3–15.0)	0.8 (0.0–15.0)	0.005 *
Screen time (hrs/day)	2.5 (1.0–6.0)	4.5 (1.5–10.0)	0.006 *

AIS: Adolescent idiopathic scoliosis. Data are presented as mean ± SD or median (range). ^1^ Data were compared using paired *t*-test or Wilcoxon matched-pairs signed rank test for non-normally distributed variables. * *p* < 0.05.

**Table 4 children-09-01619-t004:** Biochemical characteristics of control and AIS participants.

	Number of Pairs (n)	Controls	AIS	*p* Value ^1^
Glucose (mmol/L)	19	4.9 (4.4–5.5)	5.1 (4.5–6.3)	0.160
Insulin (pmol/mL)	14	58.11 ± 20.14	60.01 ± 23.88	0.755
HOMA-IR	14	1.8 ± 0.7	2.0 ± 1.0	0.584
C-peptide (pg/mL)	17	1101 ± 287	1181 ± 295	0.348
QUICKI	14	0.35 ± 0.02	0.35 ± 0.03	0.615
Triglycerides (mmol/L)	19	0.75 (0.40–1.22)	0.70 (0.49–1.42)	0.868
Cholesterol (mmol/L)	19	3.78 ± 0.49	4.06 ± 0.63	0.111
HDL-C (mmol/L)	19	1.35 ± 0.20	1.36 ± 0.21	0.854
LDL-C (mmol/L)	19	2.07 ± 0.40	2.35 ± 0.53	0.063
ApoB (g/L)	19	0.63 ± 0.13	0.71 ± 0.15	0.108
ApoA1	18	1.37 (1.11–1.73)	1.42 (1.19–1.82)	0.502
ApoB/ApoA1	18	0.46 ± 0.10	0.50 ± 0.11	0.243

HOMA-IR: Homeostatic Model Assessment of Insulin Resistance; QUICKI: Quantitative insulin sensitivity check index; HDL-C: High-density lipoprotein-cholesterol; LDL-C: Low-density lipoprotein-cholesterol; ApoB: Apolipoprotein B; ApoA1: Apolipoprotein A1; AIS: Adolescent idiopathic scoliosis. Data are presented as mean ± SD or median (range). ^1^ Data are compared using paired *t*-test or Wilcoxon matched-pairs signed rank test for non-normally distributed variables.

**Table 5 children-09-01619-t005:** Fasting levels of metabolic parameters in control and AIS participants.

	Number of Pairs (n)	Controls (n = 19)	AIS (n = 19)	*p* Value ^1^
Leptin (ng/mL)	16	4.6 (1.3–9.8)	2.8 (0.4–10.3)	0.083
Adiponectin (μg/mL)	15	26.2 (6.6–73.8)	84.7 (11.1–287.2)	0.007 *
Leptin/adiponectin ratio	12	0.258 (0.024–1.053)	0.042 (0.005–0.320)	0.005 *
Resistin (ng/mL)	19	14.29 (9.52–19.98)	14.90 (10.15–42.29)	0.358
Visfatin (ng/mL)	13	5.93 (2.40–61.50)	4.12 (1.00–34.46)	0.787
Active ghrelin (pg/mL)	13	66.68 ± 39.48	81.10 ± 51.06	0.357
Total ghrelin (pg/mL)	19	384.5 ± 133.8	452.3 ± 265.6	0.160
GIP (ng/mL)	17	366.8 (149.4–660.6)	415.2 (136.4–1408)	0.182
GLP-1 active (pmol/L)	17	7.59 (5.95–18.71)	7.63 (6.08–9.06)	0.678
GLP-1 total (pmol/L)	19	25.33 ± 7.78	25.20 ± 10.58	0.970
GLP-2 (ng/mL)	17	1.90 ± 0.66	2.08 ± 0.65	0.460
PYY (pg/mL)	19	88.10 ± 55.99	76.38 ± 59.45	0.510
DPP-4 activity (pmol/min)	18	580.5 ± 169.0	551.8 ± 193.3	0.330
OCN total (ng/mL)	19	51.85 ± 14.83	46.62 ± 11.83	0.237
ucOCN (ng/mL)	18	10.09 (3.05–51.77)	10.49 (1.73–39.21)	0.832
ucOCN/OCN ratio (%)	18	28.22 ± 22.95	29.16 ± 19.23	0.841

GIP: Gastric inhibitory polypeptide; GLP-1: Glucagon-like peptide-1; GLP-2: Glucagon-like peptide-2; PYY: Peptide tyrosine tyrosine; DPP-4: Dipeptidyl peptidase-4; OCN: Osteocalcin; ucOCN: Uncarboxylated osteocalcin; AIS: Adolescent idiopathic scoliosis. Data are presented as mean ± SD or median (range). ^1^ Data were compared using paired *t*-test or Wilcoxon matched-pairs signed rank test for non-normally distributed variables. * *p* < 0.05.

## Data Availability

Not applicable.

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
