# Peer review of "Circulatory Adipokines and Incretins in Adolescent Idiopathic Scoliosis: A Pilot Study"

_children, 2022, doi:10.3390/children9111619_

Round 1
Reviewer 1 Report
Yes. But on the condition that the title includes a "pilot study". or the manuscript will be published under "preliminary study" or a similar section. ELISA was a good research method in the 1980s and 1990s. I myself have published a lot of works using this method. However, it is not enough now.Yes. But on the condition that the title includes a "pilot study". or the manuscript will be published under "preliminary study" or a similar section. ELISA was a good research method in the 1980s and 1990s. I myself have published a lot of works using this method. However, it is not enough now.Yes. But on the condition that the title includes a "pilot study". or the manuscript will be published under "preliminary study" or a similar section. ELISA was a good research method in the 1980s and 1990s. I myself have published a lot of works using this method. However, it is not enough now.Yes. But on the condition that the title includes a "pilot study". or the manuscript will be published under "preliminary study" or a similar section. ELISA was a good research method in the 1980s and 1990s. I myself have published a lot of works using this method. However, it is not enough now.Yes. But on the condition that the title includes a "pilot study". or the manuscript will be published under "preliminary study" or a similar section. ELISA was a good research method in the 1980s and 1990s. I myself have published a lot of works using this method. However, it is not enough now.Yes. But on the condition that the title includes a "pilot study". or the manuscript will be published under "preliminary study" or a similar section. ELISA was a good research method in the 1980s and 1990s. I myself have published a lot of works using this method. However, it is not enough now.Yes. But on the condition that the title includes a "pilot study". or the manuscript will be published under "preliminary study" or a similar section. ELISA was a good research method in the 1980s and 1990s. I myself have published a lot of works using this method. However, it is not enough now.
Reviewer 2 Report
This manuscript is a cross-sectional pilot study aimed to compare anthropometry, BMD and metabolic profile of 19 AIS girls to 19 age-matched healthy controls. Collected data included participants’ fasting metabolic profile, anthropometry (measurements and DXA scan), nutritional intake and physical activity level. AIS girls (14.8 ± 1.7 years, Cobb angle 27 ± 10°) compared to controls (14.8 ± 2.1 years) were leaner (BMI-for age z-score ± SD: -0.59 ± 0.81 vs. 0.09 ± 1.11, P=0.016; fat percentage: 24.4 ± 5.9 vs. 29.2 ± 7.2 %, P=0.036), had lower BMD (total body without head z-score ± SD: -0.6 ± 0.83 vs. 0.23 ± 0.98, P=0.038; femoral neck z-score: -0.54 ± 1.20 vs. 0.59 ± 1.59, P=0.043), but their height was similar. AIS girls had higher adiponectin levels [56 (9-287) vs. 32 (7-74) ug/ml, P=0.005] and lower leptin/adiponectin ratio [0.042 (0.005-0.320) vs. 0.258 (0.024-1.053), P=0.005].
I read the article with interest, the title is well thought out and faithfully reflects the content of the study.
The abstract is adequately developed. In the introduction, the characteristics of the adolescent idiopathic scoliosis have been described. In materials and methods are adequately developed. The discussion is sufficiently developed.
Nevertheless, some minor changes are needed to be considered suitable for publication.
Comment 1: In the Abstract: It would be appropriate to refer to the characteristics, they do not seem to be very clear
Comment 2: In the introduction: It would be better to elaborate on the aspects on the pathogenesis of the adolescent idiopathic scoliosis, adding appropriate bibliographical references. For example: (Canavese F. (2020) "Idiopathic scoliosis").
Comment 3: In the introduction: It would be appropriate to add brief notes on the treatment, adding appropriate bibliographical references. For example: (Di Maria F, et al. (2021) "Immediate Effects of Sforzesco® Bracing on Respiratory Function in Adolescents with Idiopathic Scoliosis. Healthcare (Basel)").
Comment 4: In the discussion: It's not very clear what should be done in future studies to improve the limitations of your study.
Comment 5: Finally, additional English editing is needed. The Non-Native Speakers of English Editing Certificate was not signed.
